# Use of the *Puccinia sorghi* haustorial transcriptome to identify and characterize AvrRp1-D recognized by the maize Rp1-D resistance protein

**Saet-Byul Kim** [1,2]*, **Ki-Tae Kim**[3], **Solhee In**[4,5], **Namrata Jaiswal**[6], **Gir-Won Lee**[7], **Seungmee Jung**[4], **Abigail Rogers**[6], **Libia F. Gómez-Trejo**[1], **Sujan Gautam**[1], **Matthew Helm**[6], **Hee-Kyung Ahn**[8], **Hye-Young Lee**[5,9], **Quentin D. Read**[10], **Jongchan Woo**[4], **Katerina L. Holan**[11], **Steven A. Whitham**[11], **Jonathan D. G. Jones**[8], **Doil Choi**[4], **Ralph Dean**[2], **Eunsook Park**[4], **Peter Balint-Kurti**[2,12]

1 Department of Plant Pathology and Center for Plant Science Innovation, University of Nebraska-Lincoln, Lincoln, Nebraska, United States of America, 2 Department of Entomology and Plant Pathology, North Carolina State University, Raleigh, North Carolina, United States of America, 3 Department of Agricultural Life Science, Sunchon National University, Suncheon, Korea, 4 Department of Molecular Biology, University of Wyoming, Laramie, Wyoming, United States of America, 5 Plant Immunity Research Center, Seoul National University, Seoul, Republic of Korea, 6 United States Department of Agriculture, Agricultural Research Service, Crop Production and Pest Control Research Unit, West Lafayette, Indiana, United States of America, 7 SML Genetree Co. Ltd., Seoul, Republic of Korea, 8 The Sainsbury Laboratory, University of East Anglia, Norwich, United Kingdom, 9 Department of Horticulture, Gyeongsang National University, Jinju, Republic of Korea, 10 USDA-ARS, Southeast Area, Raleigh, North Carolina, United States of America, 11 Department of Plant Pathology, Entomology, and Microbiology, Iowa State University, Ames, Iowa, United States of America, 12 Plant Science Research Unit, USDA-ARS, Raleigh, North Carolina, United States of America

* saetbyul.kim@unl.edu

**Data Availability Statement:** The raw RNA-seq read data is deposited with links to BioProject

## Abstract

The common rust disease of maize is caused by the obligate biotrophic fungus *Puccinia sorghi*. The maize *Rp1-D* allele imparts resistance against the *P. sorghi* IN2 isolate by initiating a defense response that includes a rapid localized programmed cell death process, the hypersensitive response (HR). In this study, to identify AvrRp1-D from *P. sorghi* IN2, we employed the isolation of haustoria, facilitated by a biotin-streptavidin interaction, as a powerful approach. This method proves particularly advantageous in cases where the genome information for the fungal pathogen is unavailable, enhancing our ability to explore and understand the molecular interactions between maize and *P. sorghi*. The haustorial transcriptome generated through this technique, in combination with bioinformatic analyses such as SignalP and TMHMM, enabled the identification of 251 candidate effectors. We ultimately identified two closely related genes, *AvrRp1-D.1* and *AvrRp1-D.2*, which triggered an *Rp1-D*-dependent defense response in *Nicotiana benthamiana*. AvrRp1-D-induced *Rp1-D*-dependent HR was further confirmed in maize protoplasts. We demonstrated that AvrRp1-D.1 interacts directly and specifically with the leucine-rich repeat (LRR) domain of Rp1-D through yeast two-hybrid assay. We also provide evidence that, in the absence of Rp1-D, AvrRp1-D.1 plays a role in

PRJNA1045081 (with sub-numbers: SRR26948106, SRR26948107, SRR26948108 and SRR26948109) at National Center for Biotechnology Information.

**Funding:** This work was supported by grants from National Institute of Food and Agriculture (NIFA) (award #2022-67013-36504) to RD, PB-K, S-BK, the United States Department of Agriculture, Agricultural Research Service (USDA-ARS) research project 5020-21220-014-00D to MH and National Science Foundation (NSF IOS-2126256) to E.P, J.C. This work was supported by the National Research Foundation (NRF) of Korea grant (2018R1A5A1023599, SRC) to E.P and D.C. The funders had no role in study design, data collection and analysis, decision to publish, or preparation of the manuscript.

**Competing interests:** The authors have declared that no competing interests exist.

suppressing the plant immune response. Our research provides valuable insights into the molecular interactions driving resistance against common rust in maize.

## Author summary

The common rust disease of maize is caused by the obligate biotrophic fungus *Puccinia sorghi*. Resistance to common rust is controlled by race-specific dominant *NLR* (nucleotide-binding domain and leucine-rich repeats) genes and by a variety of non-race-specific quantitative trait loci. The maize Rp1-D is a coiled-coil-NLR protein conferring race-specific resistance that includes a rapid localized programmed cell death, hypersensitive response (HR). In this study, to identify AvrRp1-D from an avirulent *P. sorghi* IN2, we employed the isolation of haustoria, facilitated by a biotin-streptavidin interaction, as a powerful approach. This method proves particularly advantageous in cases where the genome information for the fungal pathogen is unavailable, enhancing our ability to explore and understand the molecular interactions between maize and *P. sorghi*. The haustorial transcriptome is generated through this technique in combination with bioinformatic analyses. We identified two closely related genes, *AvrRp1-D.1* and *AvrRp1-D.2*, which triggered an *Rp1-D*-dependent defense response in *Nicotiana benthamiana*. AvrRp1-D-induced Rp1-D-dependent HR was further confirmed in maize protoplasts. We demonstrated that AvrRp1-D.1 interacts directly and specifically with the leucine-rich repeat domain of Rp1-D through yeast two-hybrid assay. Our research provides valuable insights into the molecular interactions driving resistance against common rust in maize.

## Introduction

In plants, pathogen recognition occurs via two broadly defined and intimately connected systems, pattern-triggered immunity (PTI) and effector-triggered immunity (ETI). The PTI response is triggered by recognition of microbe-derived molecules known as microbe- or pathogen-associated molecular patterns (PAMPs) by receptors located in the plant cell membrane known as pattern recognition receptors (PRRs). ETI is mediated by intracellular resistance (R)-proteins. Pathogens secrete a class of molecules (usually proteins), known as effectors, into the host cell or apoplast to enhance the pathogenesis process. In adapted pathogens, a subset of these effectors suppresses host PTI and allows infection. R-proteins detect the presence of specific effectors, known as avirulence (Avr) proteins, either through direct interaction or via the action of the effectors on another molecule that is monitored (or 'guarded') by the R-protein [1]. Effector detection triggers the ETI defense response that includes many of the same responses noted for PTI as well as, often, a "hypersensitive response" (HR), a rapid localized cell death response at the point of pathogen penetration [2]. The ETI and PTI responses are integrated in ways that are not entirely understood, such that each activation appears to potentiate the other response [3, 4].

Most *R*-genes encode cytoplasmic proteins carrying NLRs (nucleotide-binding domain and leucine-rich repeats). Most NLRs carry either coiled-coil (CC) or Toll-interleukin receptor (TIR) domains at their N-termini. In general, the N-terminal domains appear to be responsible for activating cell death pathways, while the NB-ARC and LRR domains are generally associated with regulating the activity of the R-protein [5–7].

Most bacterial pathogen effectors can be identified by the presence of a type-III secretion signal [8], while oomycete effectors often possess conserved domains such as the RxLR motif that are also believed to mediate translocation into the host cell [9]. Fungal effectors do not possess comparable identifying features and instead have been identified using a combination of bioinformatic and differential expression approaches. Most biotrophic fungi, including rusts, produce infection structures called haustoria that penetrate the host cell wall, invaginating the host plasma membrane, forming an intimate connection through which molecules are exchanged in both directions. Most effectors are believed to be expressed in haustoria [10] and introduced into the host cell via these structures. Haustorial transcriptomes have therefore often been used as a starting point for the bioinformatic detection of effector candidates [10, 11]. Other criteria used to identify candidate effectors include small size, high cysteine content, potential secretion signals, and taxonomical specificity. A number of specialized tools have been developed for this purpose [12].

As obligate biotrophs, rust fungi are difficult to manipulate, which has constrained research on identifying and characterizing their effectors and *Avr* genes. Nevertheless, significant progress has been made, especially since genomes of several rust fungi have become available [13]. In maize, two *Avr*-genes have been identified from *P. polysora*, a causal agent of southern corn rust. *AvrRppC* was cloned by sequencing avirulent isolates against the corresponding resistance gene *RppC* [14]. Another gene, *AvrRppK* was identified through the co-expression with *RppK* in maize protoplasts and transgenic plant carrying *AvrRppK* lacking its signal peptide [15]. Co-expression of almost all of these cloned rust *Avr* genes with their corresponding *R*-genes in *Nicotiana benthamiana* triggers HR cell death [10, 16–19].

Resistance to maize common rust (CR), caused by the obligate biotrophic fungus *P. sorghi*, is controlled by race-specific single dominant NLR genes termed *Rp* genes [20] and by a variety of non-race-specific quantitative trait loci (QTL) [21]. CR resistance conferred by *Rp* genes is generally associated with HR. Over 25 *Rp* genes have been identified in maize, fourteen of which were mapped to the *Rp1* locus (*Rp1-A* to *Rp1-N*) [22]. Subsequently, it was reported that some of these specificities are likely closely linked rather than allelic [23, 24]. Rp1-D is a CC-NLR protein conferring race-specific HR-mediated resistance [25, 26]. Due to its complex nature, the *Rp1* locus is highly unstable and recombinogenic [24]. The *Rp1-D* haplotype carries *Rp1-D* and eight paralogues, called *rp1-dp1* to *rp1-dp8*, seven of which encode full NLR proteins [27]. A number of novel alleles derived from intragenic recombination between homologs at the *Rp1-D* locus have been identified [28]. *Rp1-D21*, an autoactive allele that causes spontaneous HR in maize and *N. benthamiana* in the absence of pathogens [5, 29, 30], is a chimeric gene derived from intragenic recombination between *rp1-dp2* and *Rp1-D* [27, 30].

In the present study, we generated a transcriptome from *P. sorghi* haustoria and, using a combination of biochemical and functional approaches, identified the effector responsible for the activation of Rp1-D, which we termed a putative AvrRp1-D. We have further characterized aspects of the AvrRp1-D/Rp1-D interaction. Among other things, we identified the regions of both proteins that are important for the specific interaction and showed AvrRp1-D functions to suppress the plant immune response when Rp1-D is absent.

## Results

### Assembly of the *P. sorghi* haustorial transcriptome

Consistent with our previous study, we observed that infection with *P. sorghi* isolate IN2 induced HR on the infected leaves of the H95: Rp1-D maize genotype, while H95 and W22 genotypes showed disease symptoms characterized by pustule formation at 5 days post-inoculation (dpi) (Fig 1A). Based on these observations, we hypothesized that *P. sorghi* IN2 produces

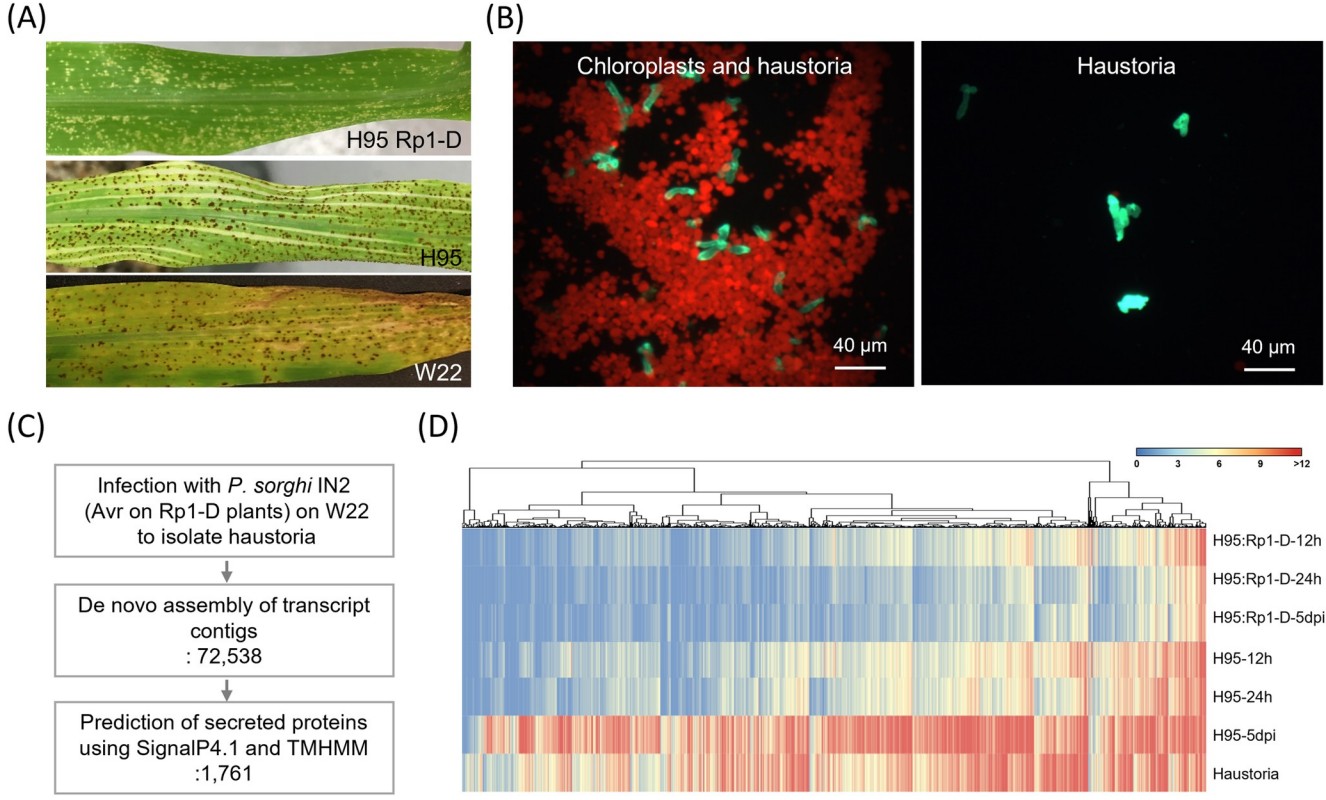

**Fig 1. Isolation of *Puccinia sorghi* IN2 haustoria and *de novo* assembly of transcripts expressed in haustoria.** (A) Hypersensitive response on a leaf of the maize line H95:Rp1-D infected with an avirulent *P. sorghi* IN2 and disease symptoms on the susceptible maize lines H95 and W22 infected with the same isolate. The photos were taken at 6-day post inoculation (dpi). (B) A mixture of chloroplasts and haustoria imaged during the haustoria purification process. Haustoria appear green due to concanavalin-A conjugated with GFP that is specifically bound to them. Chloroplasts appear red due to autofluorescence. (C) Overview of the process used to identify effector candidates expressed in haustoria of *P. sorghi* IN2 and their screening in *N. benthamiana*. (D) The heatmap shows the expression pattern of 1,761 secreted proteins, hierarchically clustered using one minus Pearson correlation and complete linkage method. Read counts of each gene were used for the heatmap. Blue and red colors indicate expression levels. The red color indicates a higher expression.

an avirulence protein recognized by Rp1-D, which we named AvrRp1-D, and that the recognition of AvrRp1-D by Rp1-D activates defense responses and HR.

Since most rust *Avr* genes are expressed in haustoria, we decided to characterize the haustorial transcriptome. Haustoria were isolated from severely infected W22 leaves at 4 dpi, before sporulation, utilizing a procedure involving biotinylated concanavalin A and streptavidin (Fig 1B) and following RNA isolation and transcriptome sequencing as detailed in the methods section.

In a previous study, we conducted deep RNA sequencing of a time course of leaves of the near-isogenic maize lines H95 (susceptible) and H95:Rp1-D (resistant) infected with *P. sorghi* IN2 [31]. Examining these data, we found that 0.10–0.66% of the derived contigs mapped to *P. sorghi* genome in H95:Rp1-D and 0.08–44.03% in H95. To investigate the variation within the fungal RNA-seq dataset, we combined the RNA-seq data conducted on H95:Rp1-D and H95 lines at 12, 24, and 120 hours, along with the haustoria samples. After removing sequences matching maize from the dataset, we determined that at 12 and 24 hours, only ∼1% of the data was of fungal origin. However, by 120 hours, over 30% of the data was of fungal origin, reflecting the increase in fungal biomass in the tissue during this time course. A principal component analysis (PCA) was then performed, demonstrating that the three or four biological replicates from each sample were well clustered, thus confirming the robustness of the dataset

(S1 Fig). *De novo* assembly identified 72,538 transcript contigs (S1 Data). We used these data to narrow down our *AvrRp1-D* candidate gene set, as explained below. While 72,538 effector genes were predicted from the haustorial sample, fewer were predicted from samples derived from infected tissues (22,747, 21,932, and 49,433 at 12, 24, and 120 hpi, respectively).

## Identification of AvrRp1-D candidates from *P. sorghi* IN2

To narrow down the list of 72,538 transcript contigs that might function as AvrRp1-D candidates, we used SignalP4.1 and TMHMM v2.0 (S1 Table). After filtering, 2,687 transcript contigs were identified. Among them, we excluded 926 transcript contigs that were expressed at 0 dpi, which might be maize genes, which left a total of 1,761 predicted proteins as potential effector candidates (Fig 1C). To further refine our search for *AvrRp1-D* candidate genes, we referred to previously generated RNA-seq data from *P. sorghi* infected H95 maize lines [31]. We reasoned that effectors would be highly expressed in H95 lines at 12 and 24 hours after infection and enriched in haustoria (Fig 1D). Based on the transcript expression data, we identified the top 251 fungal gene transcripts that were most highly expressed on average over the two-time points as potential effector candidates (S2 Table).

## High-throughput screening of effectors in *N. benthamiana* identifies an *AvrRp1-D* candidate

It has been reported that Rp1-D primarily localizes in the plant cell cytosol when transiently expressed in *N. benthamiana* [6]. Many previous studies have demonstrated that co-expression of wheat NLRs with their cognate rust Avr proteins in *N. benthamiana* causes HR [14, 15, 32, 33]. We hypothesized that this might also be the case for Rp1-D and the putative cognate effector. Therefore, we cloned effector candidates lacking their signal peptide sequence into a potato virus X (PVX) vector optimized for *in planta* expression in *N. benthamiana* [34]. All clones were sequenced, and the sequencing results revealed that 9 clones represented alternative spliced transcripts with early stop codons. The sequence analysis indicated that, of the 241 *de novo* assembled sequences, 202 showed approximately 94% or higher identity with the original putative effector sequences. Consequently, we successfully transformed 241 effector candidates into *Agrobacterium* GV3101 for subsequent experiments.

For high-throughput screening of the functional *AvrRp1-D*, agrobacterium carrying an *Rp1-D* expression construct was infiltrated using a needless syringe into entire leaves of *N. benthamiana*. Subsequently, agrobacterium containing PVX vectors carrying genes coding for the effector candidates were introduced into these *Rp1-D* infiltrated leaves at 1 dpi using a toothpick to prick the leaves at specific points (S2 Fig). Transient expression of 48 of the 241 effector candidates induced cell death in the leaf when co-expressed with *Rp1-D*. However, we considered the possibility that these effectors might cause *Rp1-D*-independent cell death or be affected by viral proteins produced in the PVX expression system [34]. To address these possibilities, we re-cloned the 48 effector candidates into a binary vector, pGWB17, driven by the 35S promoter. These candidates were then co-expressed with and without *Rp1-D* using agroinfiltration in *N. benthamiana*. In these experiments, two effector candidates, 4A12 and 597, encoding proteins of unknown function, induced *Rp1-D* independent cell death (S3 Fig). One candidate, *3E3*, caused robust and consistent cell death in *N. benthamiana* when co-expressed with *Rp1-D* but not when expressed alone. By aligning the cloned gene sequence with the *de novo* assembled sequence of *3E3*, we noted that our original *3E3* clone constituted a partial transcript. We determined the 3' region of the complete *3E3* gene using 3' RACE and showed that the full-length 3E3 lacking signal peptide also induced cell death in *N. benthamiana*

dependent on the co-expression of Rp1-D (Fig 2A and 2C). Based on this data, *3E3* was designated as a putative *AvrRp1-D*.

While cloning the full-length *AvrRp1-D* gene from the cDNA of H95 infected with *P. sorghi* IN2, we also cloned a variant of *AvrRp1-D.1*, *AvrRp1-D.2*, which encoded a protein with a 21-amino acid deletion at the N-terminus and five non-synonymous changes when compared to AvrRp1-D.1. AvrRp1-D.2 induced Rp1-D-dependent cell death in *N. benthamiana* (S4 Fig). The genomic sequences of the *AvrRp1-D.1* and *AvrRp1-D.2* genes each comprise four exons with the entire gene spanning 3026 bp in each case. *AvrRp1-D.1* (NCBI# OR593746) and *AvrRp1-D.2* (NCBI# OR593747) encodes 912 and 891 amino acids, respectively. When they were blasted in NCBI, no similarity to any proteins from other fungal species was found. AvrRp1-D is predicted to have two nuclear localization signals (NLS), one near the N-terminus and one near the C-terminus. As both *AvrRp1-D.1* and *AvrRp1-D.2* induced the same *Rp1-D*-dependent cell death and encoded almost identical proteins, we decided to use *AvrRp1-D.1* for further experiments.

## Expression of *AvrRp1-D.1* in maize also causes *Rp1-D*-specific cell death

To verify that *AvrRp1-D.1* also conferred *Rp1-D*-dependent cell death in maize, we used a protoplast system. *Rp1-D* and *AvrRp1-D.1* genes were co-transfected with the firefly luciferase gene into protoplasts isolated from the susceptible maize line H95. Co-expression of *AvrRp1-D.1* with *Rp1-D* and luciferase led to a reduction in luminescence, indicative of cell death, while *AvrRp1-D.1* or *Rp1-D* alone co-expressed with luciferase did not elicit a similar response (Fig 2D). This finding indicates that *AvrRp1-D.1* can induce *Rp1-D*-dependent cell death in both *N. benthamiana* and maize, suggesting a conserved interaction between the two proteins across expression in different organisms. We also observed that the full-length AvrRp1-D.1, including the signal peptide sequence of 22 amino acids, did not induce cell death in the presence of Rp1-D, indicating the intracellular expression of AvrRp1-D.1 is required for Rp1-D-dependent cell death (S5 Fig).

## AvrRp1-D.1 suppresses chitin-mediated ROS production

The production and accumulation of reactive oxygen species (ROS) is one of the earliest plant defense responses activated by pathogens [35, 36]. As such, virulent pathogens often secrete effectors either to the host apoplast or cytosol to suppress ROS production, thereby circumventing host immune responses [37]. To assess whether AvrRp1-D.1 suppresses an early defense response, we transiently expressed cMYC-tagged AvrRp1-D (without the predicted signal peptide) in *N. benthamiana* and tested its ability to suppress PAMP-triggered ROS burst. *N. benthamiana* leaf discs transiently expressing AvrRp1-D.1 were treated with chitin, and ROS production was monitored over time. We transiently expressed Super Yellow Fluorescent Protein (sYFP2) as a reference control. Transient expression of AvrRp1-D.1 in *N. benthamiana* suppressed chitin-mediated ROS accumulation by approximately 60% compared to the sYFP2 control (Fig 2E).

## Co-expression of six other *Rp1* alleles with AvrRp1-D results in little or no cell death

Sun et al. [27] identified eight additional *Rp1* alleles and *Rp1-D* at the *Rp1-D* locus on chromosome 10, designated *Rp1-dp1* through *Rp1-dp8*. All showed a high level of amino acid similarity with Rp1-D, ranging from 88% to 94%, except for Rp1-dp4, which contains an early stop codon in the NB-ARC domain (S3 Table). Of these nine alleles, only *Rp1-D* conferred race-specific resistance. We would expect therefore that our candidate *AvrRp1-D.1* would not

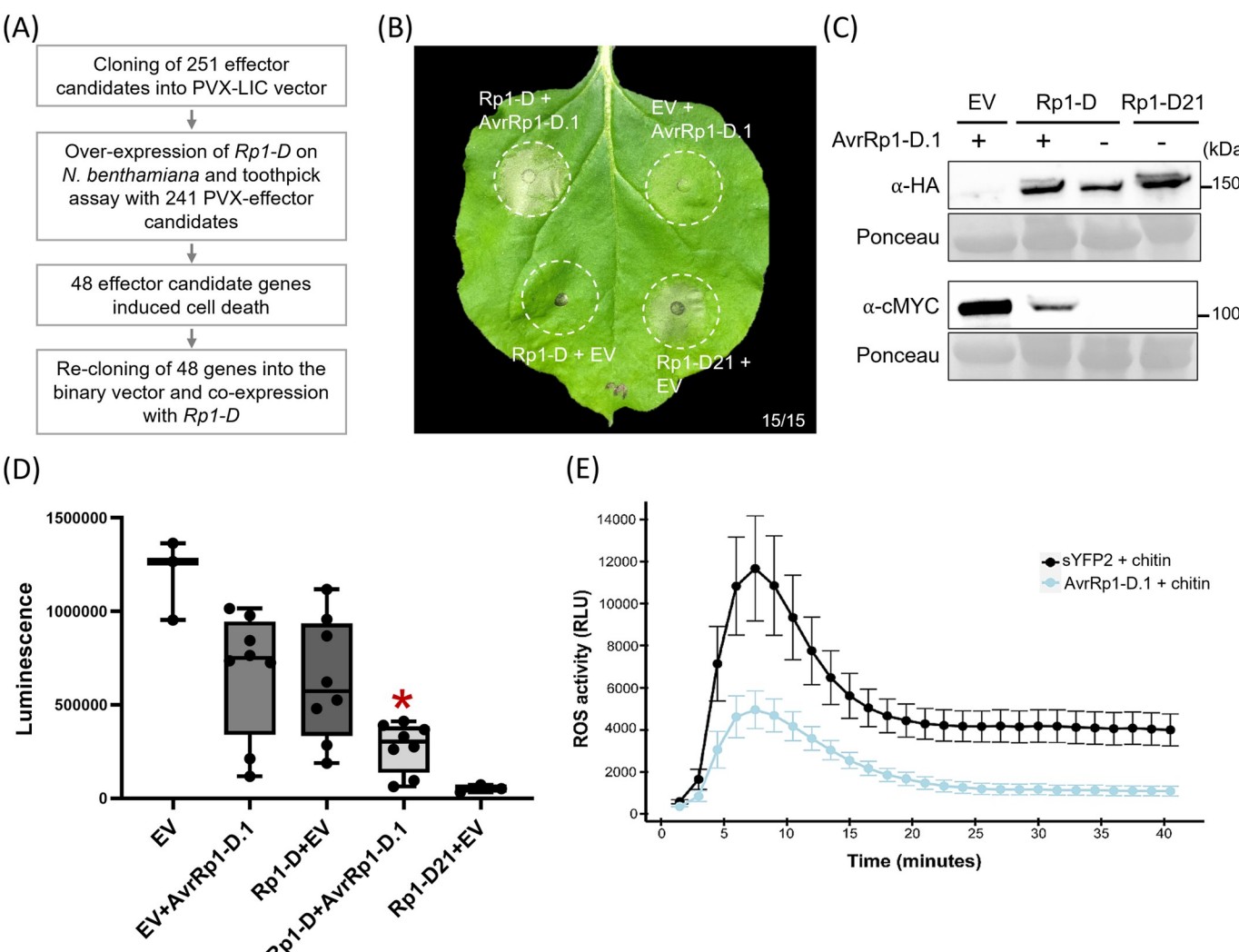

**Fig 2. AvrRp1-D.1 effector induces Rp1-D-dependent cell death and suppresses chitin-mediated ROS production.** (A) Overview of the process for identifying effector candidates conferring cell death in *N. benthamiana*. (B) Transient expression of Rp1-D:3xHA with AvrRp1-D.1:4xcMYC or empty vector (EV). HA is hemagglutinin. Rp1-D21:3xHA and empty vector were used as positive and negative controls, respectively. Representative leaf was photographed at 4 dpi. The white dashed circles indicate agroinfiltrated areas. 15 individual plants were infiltrated and showed similar results. (C) Protein expression of Rp1-D and AvrRp1-D.1 transiently expressed in *N. benthamiana*. Total protein was extracted from agro-infiltrated leaves at 36 hours post infiltration (hpi), and anti-HA or anti-cMYC antibody was used to detect the expression of the fused proteins. The sizes of the proteins are indicated on the right. Ponceau S staining of the Rubisco subunit showed equal loading of protein samples. (D) Luminescence detected in protoplasts co-expressing *Rp1-D* with *AvrRp1-D.1* or expressing each gene alone (co-transformed with an empty vector). Lower luminescence indicates increased cell death. *Rp1-D21*, an autoactive allele of *Rp1-D*, is used as a positive control for HR activation. Six independent biological replicates were tested. The asterisk indicates a significant difference between *Rp1-D*/EV and *Rp1-D*/*AvrRp1-D.1* expression. (paired *t*-test, *p<0.05). (E) *AvrRp1-D.1* suppresses chitin-mediated ROS production. The indicated constructs were transiently expressed in *N. benthamiana*. 48 hpi leaf discs were collected and challenged within chitin, and relative luminescence was monitored using a luminol-based assay. Super Yellow Fluorescent Protein (sYFP2) was used as a reference control. Error bars represent the standard error of the mean (SEM). Three independent experiments were performed with similar results.

confer cell death in *N. benthamiana* when co-expressed with any of these other seven alleles. We amplified and cloned *Rp1-dp1*, *Rp1-dp2*, *Rp1-dp5*, *Rp1-dp6*, *Rp1-dp7* and *Rp1-dp8* into expression vectors. We were not able to amplify *Rp1-dp3*. When co-expressed with AvrRp1-D.1, Rp1-D induced strong cell death as previously observed, and Rp1-dp2 induced weaker cell death, while expression of the other alleles [Rp1-dp1, 5, 6, 7, 8] with AvrRp1-D.1 did not confer any noticeable cell death (Fig 3). Likewise, AvrRp1-D.2 also triggered cell death

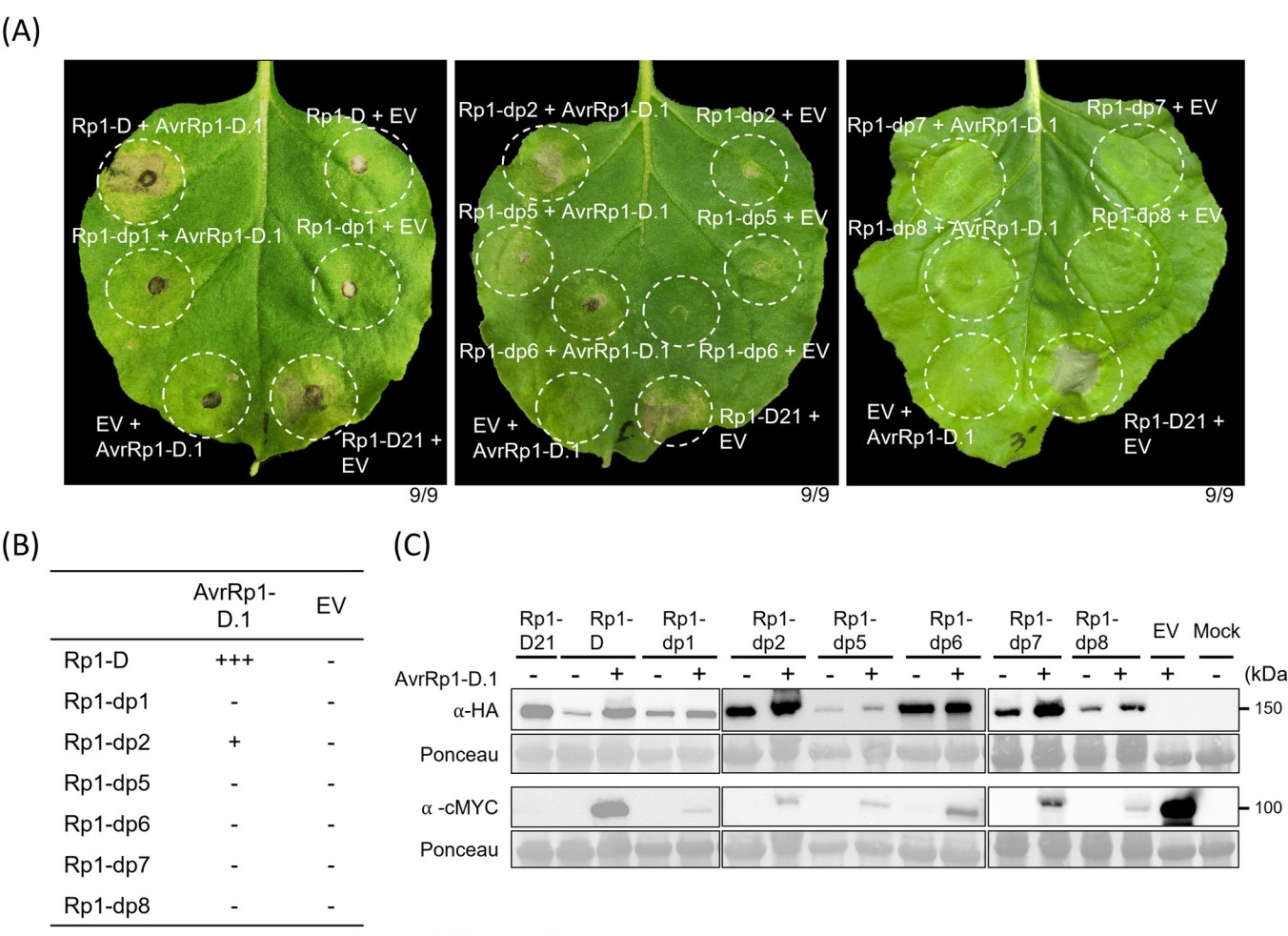

**Fig 3. AvrRp1-D.1 specifically activates the Rp1-D allele.** (A) Rp1-dp1, -dp2, -dp5, -dp6, -dp7, and -dp8 fused with a C-terminal tag 3xHA were co-infiltrated with AvrRp1-D.1 fused with a C-terminal 4xcMYC or EV in *N. benthamiana*. A representative leaf was photographed at 4 dpi. 9 individual plants were infiltrated and showed similar results. (B) The table to the right indicates the relative strength of the HR induced by the co-expression of *AvrRp1-D.1* with each *Rp1* allele (+++ is the strongest HR). Rp1-D21 was used as a positive control. (C) Protein expression of Rp1 alleles and AvrRp1-D.1. Total protein was extracted from agro-infiltrated leaves at 36 hpi, and anti-HA or anti-cMYC antibody was used to detect the expression of the fused proteins. The sizes of the proteins are indicated on the right. Ponceau S staining of the Rubisco subunit showed equal loading of protein samples.

when co-expressed with Rp1-D and, more weakly, with Rp1-dp2 (S6 Fig) but not with the other alleles.

## Analysis of Rp1-D chimeras identifies regions important for activation by AvrRp1-D

A previous study [5] demonstrated that the interaction between the NB-ARC domain and the LRR domain enables the induction of HR triggered by the auto-active Rp1-D21. This was accomplished using a series of chimeric proteins generated through domain swapping between Rp1-D and Rp1-dp2 (S7 Fig). To identify the specific domain of Rp1-D that interacts with AvrRp1-D.1, we co-expressed AvrRp1-D.1 with all these chimeric genes, which did not induce auto-active cell death (Fig 4A and 4B).

Among these, the V17 chimera, which was largely identical to Rp1-D but carried the C terminal amino acids 1090–1292 from the LRR domain of Rp1-dp2, lost the ability to induce cell death upon co-expression with AvrRp1-D.1. On the other hand, three chimeric proteins (V1,

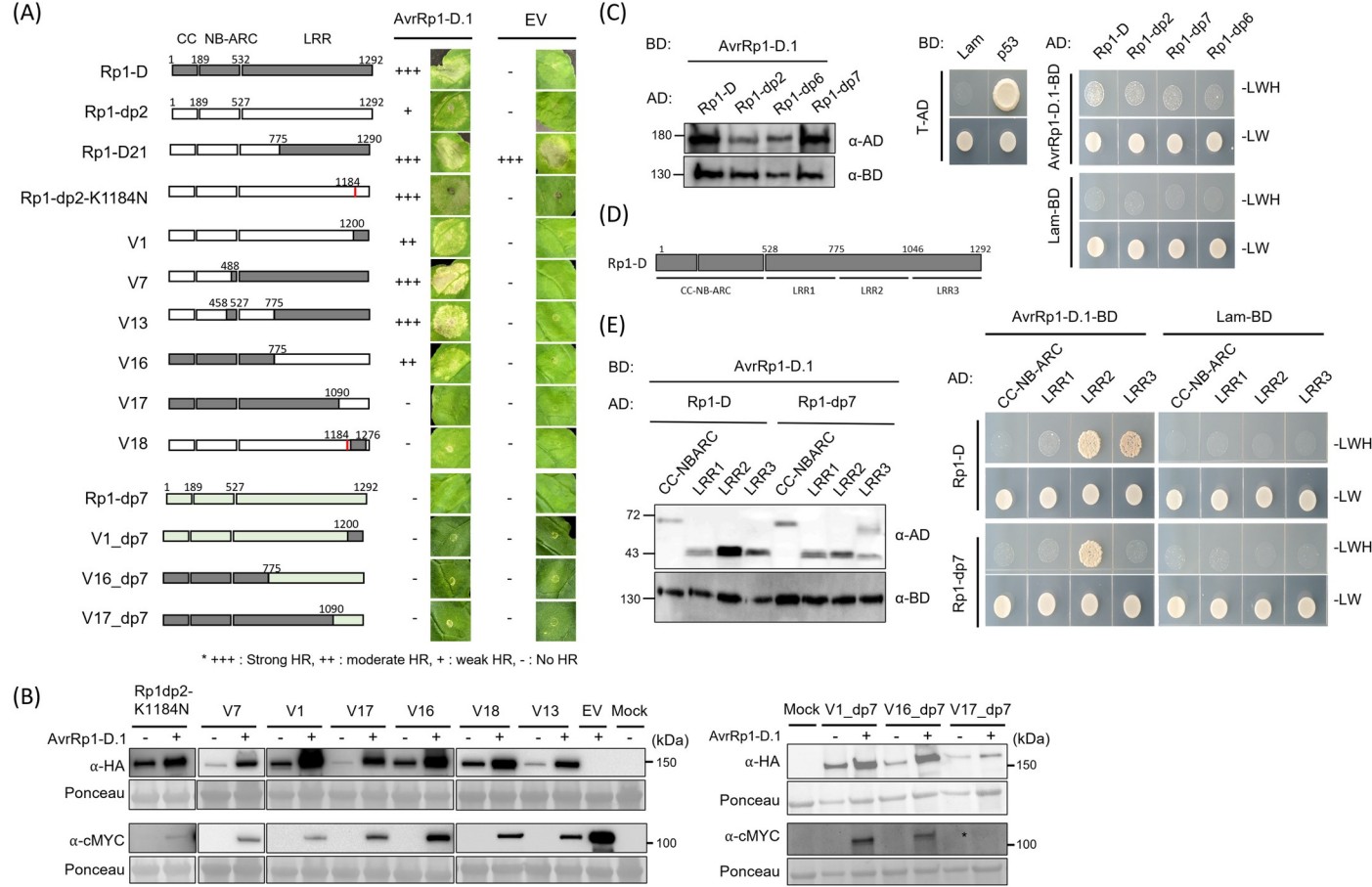

**Fig 4. Investigation of regions of Rp1-D responsible for interaction with AvrRp1-D.1.** (A)-(B) All constructs tested were described previously (5). The amino acid position of the recombination site of each construct was indicated above the construct. The red line indicates a mutated amino acid position. The domain swap constructs were fused with the C-terminal tag 3xHA, and AvrRp1-D.1 was fused with the C-terminal tag 4xcMYC. The strength of HR resulting from co-expression in *N. benthamiana* is shown. 9 individual plants were infiltrated and showed similar results. The table to the right indicates the relative strength of the HR induced by the co-expression of AvrRp1-D.1 with each Rp1 chimera or mutant (+++ is the strongest HR). Rp1-D21 was used as a positive control. (C) Yeast two-hybrid assay using strains co-expressing Rp1-D, -dp2, -dp6, and -dp7 fused to the GAL4 activation domain (AD) with AvrRp1-D.1 fused to the GAL4 DNA binding domain (BD) on control media lacking leucine and tryptophan (-LW) or selective media additionally lacking histidine (-LWH). Growth on selective media indicates protein-protein interactions. Interaction between T antigen with GAL4 activation domain and Lam or p53 with GAL4 DNA binding domain were used as negative/positive control, respectively. AD- or BD-binding proteins were detected using anti-GAL4 and anti-GAL4 (DBD). (D) Schematic diagram of the parts into which Rp1-D was divided for the yeast-two-hybrid assay shown in (E). (E) Yeast two-hybrid assay using strains co-expressing each part of Rp1-D or Rp1-dp7 shown in the schematic above fused to AD with AvrRp1-D.1 or Lam fused to BD on control media lacking leucine and tryptophan (-LW) or selective media additionally lacking histidine (-LWH). Pictures were taken 5 days after plating.

V7, and V13), containing the last portion of the LRR domain (amino acids 1200–1292) from Rp1-D, induced strong cell death in the presence of AvrRp1-D.1, similar to Rp1-D itself. Interestingly, V18, which carried part of the LRR domain (amino acids 1200–1276) from Rp1-D also did not induce cell death with AvrRp1-D.1.

Wang et al. [5] showed that when transiently expressed in *N. benthamiana*, the single K1184N mutation, which replaced the Rp1-dp2 with the Rp1-D amino acid at that spot, in construct V1 (with a combination of amino acids 1–1200 from Rp1-dp2 and amino acids 1200–1292 from Rp1-D), produced an auto-active cell death while V1 did not. This result suggested that the amino acid 1184 is important for regulating activity. Interestingly, however, Rp1-dp2 (K1184N) did not confer HR, suggesting that additional polymorphisms between Rp1-D and Rp1-dp2 in the C-terminal region are required to confer autoactivity. When AvrRp1-D.1 was co-expressed with Rp1-dp2-K1184N, it conferred stronger cell death than

when co-expressed with Rp1-dp2, providing further evidence that this specific amino acid is important for the regulation of Rp1-D activity (Fig 4A). V16, comprising the CC-NB-ARC domain and the N-terminal region of the LRR domain of Rp1-D (amino acids 1–775) and the LRR domain of Rp1-dp2 (amino acids 776–1292), caused moderate cell death when co-expressed with AvrRp1-D. The findings imply that the C-terminal LRR domains of Rp1-D likely play a role in recognizing AvrRp1-D.1. However, the fact that V16 induced a stronger Rp1-D dependent HR compared to Rp1-dp2 suggests that additional portions of Rp1-D are crucial for AvrRp1-D.1 recognition. To investigate this further, we created three chimeric mutants (V1_dp7, V16_dp7, and V17_dp7) which mirror the V1, V16 and V17 chimeras except that the portions from Rp1-dp2 are replaced with portions from Rp1-dp7 (Fig 4). Compared to V7 and V16, both V1_dp7 and V16_dp7 lost the ability to induce AvrRp1-D.1 dependent HR. These results further underline that Rp1 activation through specific recognition is a complex process involving domains throughout the protein.

Both Rp1-D and Rp1-dp2 contain so-called MHD and LHD motifs within the NB-ARC domain (S7 Fig). We previously demonstrated that the mutation of the LHD motif in Rp1-dp2 (Rp1-dp2 (D517V)) induced autoactive cell death, while other MHD and LHD mutants, including Rp1-D (H517A), Rp1-D (D518V), Rp1-D (H521A), and Rp1-D (H517AD518V), did not [5]. To determine whether the mutations are involved in AvrRp1-D.1-dependent cell death, we co-expressed the four non-autoactive MHD mutants with AvrRp1-D.1 or AvrRp1-D.2 (S8 Fig). Co-expression of Rp1-D-D518V or Rp1-D-H517AD518V with AvrRp1-D did not induce cell death, suggesting that amino acid 518 might be important for AvrRp1-D dependent activation. In summary, our experiments suggest that amino acids 1090–1292 within Rp1-D, and amino acid 1184 within Rp1-dp2 have a crucial role in the control of its activation by AvRp1-D.1.

## Rp1-D and Avr-Rp1-D.1 physically interact in an allele-specific manner

To establish the direct interaction between Rp1-D and AvrRp1-D.1, we conducted a yeast two-hybrid assay. Rp1-dp2, Rp1-dp6, and -dp7 were used as controls since they induced lower levels of cell death (Rp1-dp2) or no cell death (Rp1-dp6, Rp1-dp7) when co-expressed with AvrRp1-D.1. Unexpectedly, the full-length proteins of Rp1-D, Rp1-dp2, Rp1-dp6, and Rp1-dp7 all demonstrated weak interactions with AvrRp1-D.1 (Fig 4C) in the yeast two-hybrid system. To further explore the specific domains responsible for the interaction, we tested the individual domains of the Rp1 protein for their interactions with AvrRp1-D.1. The Rp1-D protein was divided into four fragments: CC-NB-ARC, LRR1 (amino acids 528–775), LRR2 (amino acids 776–1046), and LRR3 (amino acids 1047–1292) (Fig 4D). Interestingly, while the LRR2 domains from both Rp1-D and Rp1-dp7 showed strong interactions with AvrRp1-D.1, only the Rp1-D LRR3 domain interacted with AvrRp1-D.1 (Fig 4E). These findings suggest that the LRR3 domain of Rp1-D may be responsible for the direct recognition of AvrRp1-D.1. This is consistent with our findings detailed above that amino 1090–1292 within Rp1-D plays a crucial role in the recognition of AvrRp1-D.1.

## AvrRp1-D.1 is nuclear localized, and its N-terminal region is required for Rp1-D activation

To gain insights into which region of AvrRp1-D.1 activates Rp1-D, we generated ten deletion mutants (Fig 5A). Each deletion clone was co-expressed with Rp1-D in *N. benthamiana*. Interestingly, all the deletion clones that carried the N-terminal region amino acids 23–229, namely del-2, -3, -4, -5, and -6, triggered Rp1-D-dependent cell death at 3 dpi with del-3 inducing the strongest cell death response (Fig 5). AvrRp1-D.1 is predicted to have two nuclear localization

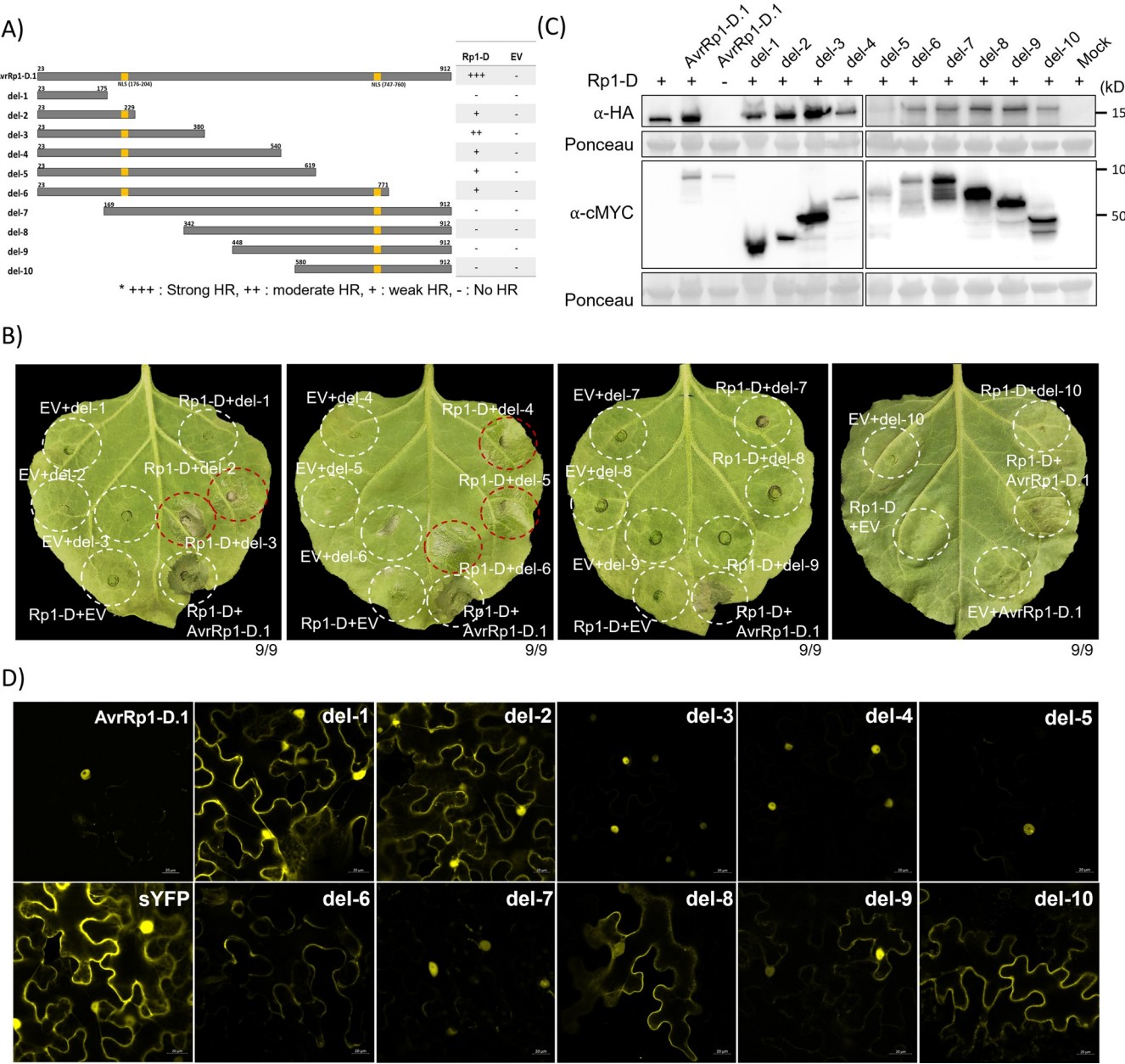

**Fig 5. The N-terminus of AvrRp1-D.1 is essential for the recognition of Rp1-D in *N. benthamiana*.** (A) Schematic representation of truncations of AvrRp1-D.1 with the corresponding amino acid positions on the right. The blue bar at the top is a schematic of the entire AvrRp1-D.1 protein and the grey bars below illustrate the portion that is expressed by each deletion construct. The table to the right indicates the relative strength of the HR induced by the co-expression of AvrRp1-D.1 with each Rp1 allele (+++ is the strongest HR). (B) Co-expression of deletion constructs of AvrRp1-D.1 with Rp1-D or EV in *N. benthamiana*. Pictures show the abaxial side of the leaves at 4 dpi. 9 individual plants were infiltrated and showed similar results. (C) Protein expression of deletion constructs of AvrRp1-D.1 and Rp1-D from experiment shown in (B). Total protein was extracted from agro-infiltrated leaves at 32 hpi, and anti-HA or anti-cMYC antibody was used to detect the expression of the fused proteins. The sizes of the proteins were labeled on the right. * indicates the target band. Ponceau S staining of the Rubisco subunit showed equal loading of protein samples. (D) The subcellular localization of truncations of AvrRp1-D.1. AvrRp1-D.1 deletion constructs fused with the C-terminal enhanced yellow fluorescent protein (EYFP) were transiently expressed in *N. benthamiana*. Confocal images were assessed at 30–36 hpi. Confocal micrographs are of single optical sections and the scale bar is 20μm.

signals (NLS) at both N-terminus and C-terminus. The co-expression results suggest that the AvrRp1-D.1 N-terminus, which includes a predicted NLS, may play a crucial role in its recognition by Rp1-D and subsequent induction of cell death.

To examine the localization of AvrRp1-D.1 and the possible role of localization in its recognition by Rp1-D, we expressed AvrRp1-D.1 and its truncated forms fused with C-terminal EYFP in *N. benthamiana*. We observed that AvrRp1-D.1 and del-3, -4, and -5 localized in the nuclei, del-2, -7, and -8 localized predominantly in the nuclei together with weak cytosolic localization, and del-1, -6, -9, and -10 localized predominantly in the cytosol (Fig 5D). Since del-6 did not localize to the nucleus, there was no clear correspondence between nuclear localization and the ability to induce HR via interaction with Rp1-D.

### The AvrRp1-D homolog from a virulent *P. sorghi* isolate also activates Rp1-D

We identified and cloned the AvrRp1-D.1 homolog from the virulent isolate IA16. We observed a single amino acid polymorphism at position 51 (Arginine to Lysine) between this and the *AvrRp1-D* allele from the avirulent IN2 isolate (S9 Fig). Somewhat unexpectedly, co-expression of the allele from isolate IA16 (which we termed AvrRp1-D$^{IA16}$) with *Rp1-D* resulted in the induction of cell death, similar to the response observed with the co-expression of *Rp1-D* and *AvrRp1-D.1* (Fig 6). To measure the expression level of the *AvrRp1-D.1* and *AvrRp1-D.2*, we performed RT-qPCR of samples from infected leaves of the susceptible maize line H95 using primers that amplified all two genes (S4 Fig). This analysis showed that the *AvrRp1-D.1* and *AvrRp1-D.2* from *P. sorghi* IN2 were expressed at approximately three-fold higher levels than the *AvrRp1-D$^{IA16}$* from *P. sorghi* IA16 (Figs 6B and S10).

## Discussion

Rp1-D is a canonical CC-NB-LRR that causes a typical HR in maize in a *P. sorghi* isolate-specific manner [26]. In this study, we first identified the effector protein AvrRp1-D.1 that activates Rp1-D using a biotin-streptavidin system for haustorial isolation. We identified regions important for this interaction in both proteins, showing that AvrRp1-D.1 functions as a suppressor of the immune response. This study also raises an interesting question concerning the reason for the existence of AvrRp1-D.1 in the virulent isolate and the interactions of Rp1-D with AvrRp1-D.1, which we address below.

### Isolation of high quality haustoria is important for "effectoromics"

The dynamic shifts in climate, coupled with the evolution of fungi, pose challenges in acquiring high-quality genome sequences. Furthermore, when faced with poor genomic quality of the pathogenic fungus, obtaining precise effector sequences becomes difficult. In biotrophic fungal pathogens, most effectors are expressed in haustoria- the main organ concerned with molecular exchange with the plant cell. In the absence of genome information, haustorial isolation may provide accurate effector sequences. There have been many attempts to isolate intact and pure haustoria from biotrophic fungi. Concanavalin A (Con A) is a lectin that binds to α-mannopyranosyl and α-glucopyranosyl residues, found in fungal cell walls [38]. Methods for haustorial isolation using concanavalin A-conjugated Sephadex column, ultracentrifugation and Fluorescence-Activated Cell Sorting (FACS) have been reported [39–41]. We were not able to isolate intact *P. sorghi* haustoria using these methods. Kunjeti et al. [42] developed a simpler method using the interaction between streptavidin and biotinylated Con A to isolate the haustoria of *Phakopsora pachyrhizi*. Using this method, we isolated intact *P. sorghi* haustoria, but not in sufficient quantities for RNA-seq. In order to isolate enough intact haustoria, the isolated haustoria mixture was incubated for an extended duration after adding the Con A-streptavidin bead complex, up to one hour longer than the previous protocol. This modification was sufficient to obtain high-quality *P. sorghi* haustorial RNA for transcriptomics.

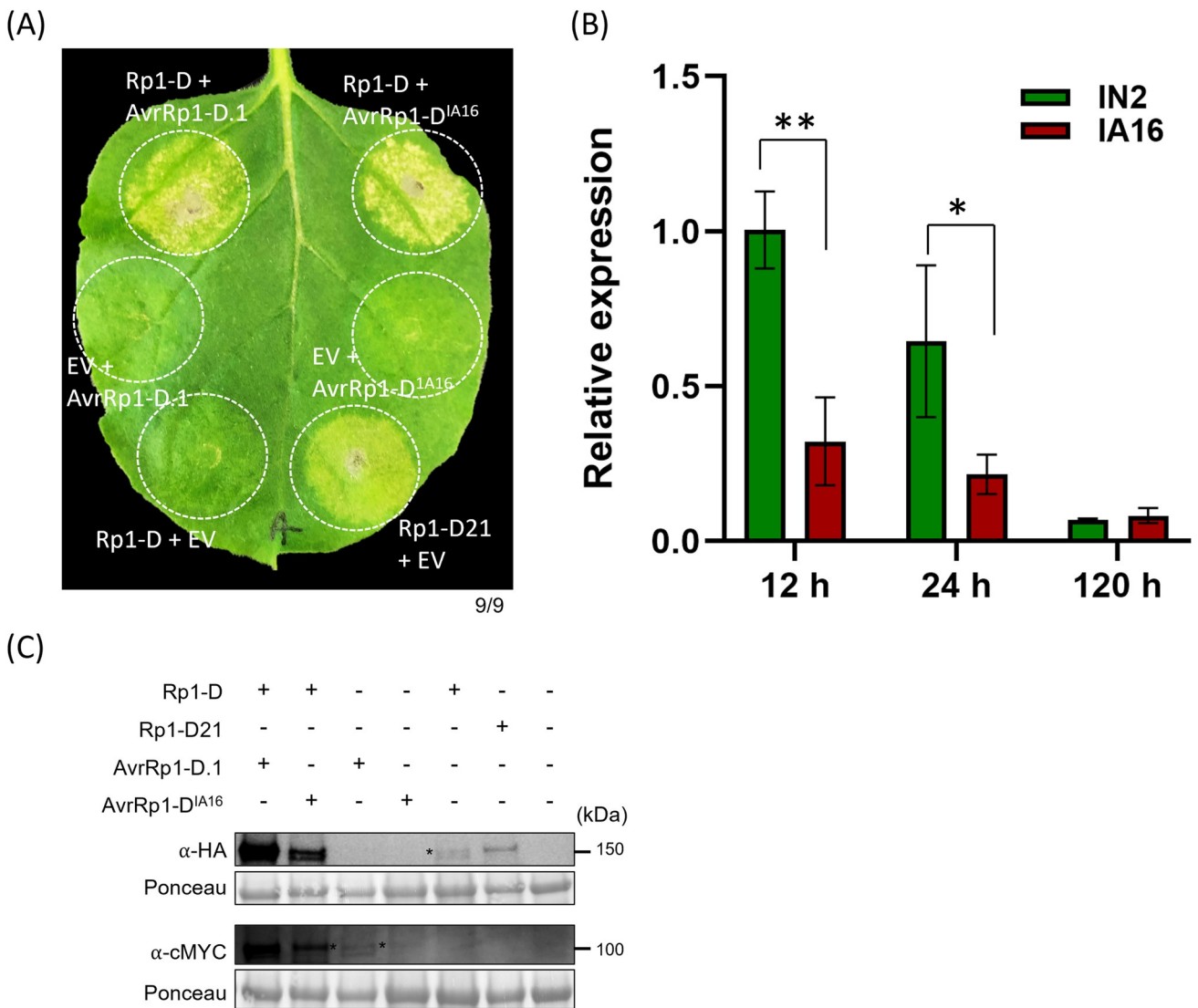

**Fig 6. The AvrRp1-D$^{IA16}$ from the virulent _P. sorghi_ isolate also activates Rp1-D.** (A) Transient expression of _AvrRp1-D$^{IA16}$_ or _AvrRp1-D.1_ with _Rp1-D_ or EV in _N. benthamiana_. A representative leaf was photographed at 6 dpi. 9 individual plants were infiltrated and showed similar results. Rp1-D21 was used as a positive control. (B) The relative expression levels of _AvrRp1-D.1_ and _AvrRp1-D.2_ from the avirulent _P. sorghi_ IN2 and the virulent _P. sorghi_ IA16 in the susceptible maize line H95 plants at 12, 24 and 120 hr (paired _t_-test, *p<0.05, **p<0.01). One biological repeat was used for the graph. Similar results were obtained in the second repeat and are shown in S10 Fig. (C) Protein expression of AvrRp1-D.1 variants and Rp1-D from experiment shown in (A). Total protein was extracted from agro-infiltrated leaves at 36 hpi, and anti-HA or anti-cMYC antibody was used to detect the expression of the fused proteins. The sizes of the proteins were labeled on the right. * indicates the target band. Ponceau S staining of the Rubisco subunit showed equal loading of protein samples.

The transcriptomic data generated from four sets of haustoria was utilized to create _de novo_ assembled sequences that were almost the same (about 94% at the nucleotide level) as the sequences derived from cDNA amplification (S2 Table). This modified method may be applicable across various rust types.

### _AvrRp1-D.1_ is a relatively large avirulence gene encoding a 912 amino acid protein

Most effectors, including most cloned avirulence genes encode small, cysteine-rich secreted proteins, often less than 300 amino acids in length. In this respect AvrRp1-D is unusual; it is

larger, has 912 amino acids, and shows no significant similarity to proteins found in other species within existing databases. A recently identified avirulence gene *AvrStb9* from *Zymoseptoria tritici*, a causal agent of septoria leaf blotch of wheat encodes a protein of 727 amino acids featuring a protease domain [43]. To the best of our knowledge, we are unaware of any reported avirulence proteins approaching this size. Given the remarkable size of both AvrRp1-D.1 and AvrStb9, in the future, it might be worth reconsidering whether and how to use size as a criterion for identifying potential effectors from haustorial transcriptomics data.

## AvrRp1-D.1 localizes in the nucleus when transiently expressed in *N. benthamiana*

In most cases where both the R and Avr components have been identified, rust R proteins have been found to interact directly with their cognate Avr proteins [32, 44]. This appears to be the situation with Rp1-D and AvrRp1-D.1. Our microscopic data suggests that AvrRp1-D.1 is primarily localized in the nucleus (Fig 5) and the nuclear localization of AvrRp1-D.1 might not be essential to induce Rp1-D-mediated HR. In contrast, previous work suggests that Rp1-D primarily located in the cytosol, requires both nuclear and cytoplasmic localization to function [6]. Further work is required to determine where the Rp1-D/AvrRp1-D interaction occurs. It is possible that Rp1-D is activated by AvrRp1-D.1 in the cytosol, while AvrRp1-D.1 itself translocates to the plant cell nucleus, potentially playing a role in gene regulation.

We and others have previously shown that an autoactive derivative of Rp1-D, Rp1-D21, directly interacts with a number of maize proteins, including proteins involved in defense-associated secondary metabolism [45–47] and components of the protein sorting and degradation pathways [48, 49]. Additionally, several studies have suggested that the localization of Rp1-D21 within the cell can be affected by the presence of interacting proteins and is important for its activity [45, 48, 49]. The picture that emerges is of Rp1-D as part of a large protein complex, distributed among various subcellular compartments depending on activation status and which other proteins it is associated with. We hope that the identification of AvrRp1-D.1 will enable studies to clarify how Rp1-D activity is controlled *in vivo*.

## Why does the *P. sorghi* isolate virulent on plants carrying *Rp1-D* carry an allele of AvrRp1-D that is capable of activating Rp1-D?

In this study, we identified the effector 3E3 from the avirulent *P. sorghi* IN2 and showed that it elicited Rp1-D-dependent HR in both *N. benthamiana* and maize. Furthermore, we demonstrated that 246 amino acids of the C-terminal section of the LRR domain of Rp1-D directly interact with 3E3 in a yeast two-hybrid system while the equivalent portions from another Rp1 allele, Rp1-dp7 does not. Based on these findings, we designated 3E3 as AvrRp1-D.1.

We also identified AvrRp1-D.2 from IN2 which differs from AvrRp1-D.1 by a 21-amino acid deletion at the N-terminus and five non-synonymous changes. It is not clear at the moment if AvrRp1-D.2 is an alternative allele of AvrRp1-D or if it is a homolog encoded at a different genomic locus. AvrRp1-D.2 appears to act in an identical way to AvrRp1-D with respect to its ability to activate Rp1-D dependent HR and to interact with Rp1-D paralogues (S4, S6 and S8 Figs).

While *AvrRp1-D.1* was isolated from *P. sorghi* isolate IN2 which is avirulent on maize lines carrying *Rp1-D*, we also identified an *AvrRp1-D* allele, *AvrRp1-D*[IA16], from the *P. sorghi* IA16 isolate, which is virulent on maize lines carrying *Rp1-D*. AvrRp1-D[IA16] differed from AvrRp1-D only at one amino acid (S9 Fig) and it induced *Rp1-D*-dependent HR in *N. benthamiana* of similar strength to AvrRp1-D.1. Upadhyaya et al. (33) reported that the Sr27 resistance protein in wheat against stem rust caused by *P. graminis* f. sp. *tritici*, is activated in

both *N. benthamiana* and wheat protoplast assays by two variants of the AvrSr27 protein (AvrSr27-1 and AvrSr27-2) from the avirulent isolate as well as one, avrSr27, from the virulent isolate, which displayed 17 amino acid differences compared to AvrSr27-1. It was determined that, during infection, *avrSr27* was expressed at 15% of the levels of *AvrSr27-1*. This suggested that the virulence determinant in this case was the expression level rather than the amino acid sequence. Similarly, this may be the case here as the expression level of *AvrRp1-D* from the avirulent isolate IN2 appears higher than that of the virulent isolate IA16 during early infection (Fig 6B).

In conclusion, we have identified AvrRp1-D by high-throughput screening of effectors in *N. benthamiana*. In the absence of Rp1-D, AvrRp1-D.1 appears to suppress the plant immune response. AvrRp1-D.1 activates Rp1-D but does not activate proteins encoded by six highly homologous alternative *Rp1* alleles, and it appears this activation specificity resides in the C terminal region of the LRR domain of Rp1-D, consequently inducing hypersensitive response (HR) in maize.

## Materials and methods

### Plant growth and *P. sorghi* inoculation

The maize line H95:Rp1-D is near-isogenic to the commonly used maize line H95 with the addition that it carries the *Rp1-D* resistance gene in a homozygous state. W22 is a commonly used maize line that is susceptible to *P. sorghi* IN2. H95:Rp1-D, H95, and W22 seedlings were grown under a 16h/8h photoperiod cycle at day/night temperatures of 26/22 and relative humidity of 60% in growth chambers in the NCSU phytotron. Two-week-old maize plants at the fourth and fifth leaf stage were inoculated in the following way: 100 μL of freshly collected spores by shaking the leaves of H95 infected by *P. sorghi* IN2 at 6–7 dpi was mixed with 900 μL of talc and rubbed using a thumb and index finger onto the fourth and fifth leaves. Inoculated seedlings were placed at 100% humidity in a plastic tent for 16 hours, then returned to the original conditions: 16h/8h photoperiod, day/night temperatures of 26/22, and a relative humidity of 60%.

*Nicotiana benthamiana* was grown for 3-to-5 weeks under a 16h/8h photoperiod cycle at day/night temperatures 25/22 and relative humidity of 60% in growth chambers or grown at room temperature with light intensity between 120–140 μmol/m$^2$/s$^1$.

### Isolation of haustoria of *P. sorghi* IN2

In a previous study, streptavidin and biotinylated concanavalin A were used to isolate *P. pachyrhizi* haustoria from infected soybean leaves [42]. We slightly modified the protocol for *P. sorghi* haustoria isolation from infected maize leaves. For example, we used 35 μm nylon mesh instead of the 25 μm nylon mesh used in the previous paper and incubated the Con-A-streptavidin bead with haustoria mixture longer up to one hour. 40g of infected leaves of W22 infected with *P. sorghi* IN2 were washed with deionized water, and the leaves were cut into small (1–3 cm) pieces. They were homogenized with 240 ml of homogenization buffer (0.3 M sorbitol, 20 mM MOPS, 1 mM DTT, 0.2 PVP, 0.1% BSA, and 0.2% RNA protect solution, adjusted pH 7.2) using a blender (https://blend-works.net/product/allinoneblender/). The macerate was passed gradually through a 100 μm nylon mesh (Linear Bifurcation Analysis) into 50 ml tubes on ice. The supernatant was passed through 35 μm nylon mesh (Linear Bifurcation Analysis) into 50 ml tubes on ice. Then, the supernatant was centrifuged for 7 minutes at 1,000 x g at 4°C to sediment the chloroplast and haustoria mixture as a green plus white pellet. We discarded the supernatant and gently broke the pellet by finger tapping. The pellet was resuspended with a suspension buffer (0.3 M sorbitol, 10 mM MOPS, 1 mM CaCl$_2$, 1 mM MnCl$_2$, 0.2% BSA) by gentle pipetting. The resuspension was divided into 10 aliquots of 1.5 ml

tubes each. 200 μl of Con-A-streptavidin bead complex (2ml containing washed streptavidin magnetic beads (NEB, S1420S) 600 μl, biotinylated concanavalin A (Sigma, C2272) 600 μl and the suspension buffer 800 μl) was added to each 1 ml aliquot of the resuspended homogenate and was kept at 4˚C for up to one hour. The mixture was then washed by placing tubes in magnetic stands and repeated two times. After removing the last wash, the beads were suspended in 500 μl of TRIzol and transferred to a glass dounce (7ml). Total RNA was isolated according to the manufacturer's protocol. To confirm the integrity of the isolated haustoria, we incubated them with FITC-labeled concanavalin A, a lectin that binds to fungal cell wall glycoproteins (Biotium, #29016, San Francisco, USA). This allowed us to observe the haustoria (green) in conjunction with chloroplasts (red).

## Prediction and annotation of genes encoding secreted proteins

Total RNA samples isolated from four independent haustoria were sequenced in 150 paired-end mode on an Illumina Novaseq platform. 72,538 transcript contigs were *de novo* assembled using Trinity software [50]. The quality check of the raw reads data was performed using Fastp v0.23.2 (DOI: 10.1093/bioinformatics/bty560), and the raw reads were mapped to the *de novo* assembled transcripts using Hisat2 v2.2.1 [51]. Next, Samtools v1.15.1 [52] was used to sort the read alignment data, and StringTie v2.2.1 was used for read counting [53]. Finally, the differentially expressed genes (DEGs) were identified by the DESeq2 R package [54] using the read count matrix derived from the prepDE program in the StringTie package [53]. The proteins translated from the ORFs were annotated using various tools to predict potential effector candidates (S2 Table). SignalP4.1 identified the signal peptide (SP) with the *default* cutoff of 0.45 [55]. SignalP 4.1 was first applied to 72,538 transcript contigs to identify effector candidates carrying signal peptides. TMHMM v2.0 was subsequently utilized to sort out the effector proteins without transmembrane domains, using the default threshold of the tool [56]. As a result, we identified 1,761 effector candidates that have signal peptides but do not have a transmembrane domain. The other tools for assisting the annotation included were ApoplastP [57], TargetP-2.0 [58], EffectorP v2.0 [59], and EffectorP v3.0 with the default cutoff [60]. BlastP [61] was used to detect maize homologs using the best hit in the 1e-5 value (Zm-B73-REFERENCE-NAM-5.0_Zm00001eb.1).

## RT-qPCR

Five micrograms of total RNA were used for first-strand cDNA synthesis using SuperScript IV reverse transcriptase (Invitrogen). A real-time RT-qPCR assay was performed on a CFX Duet Real-Time PCR System using 2X SYBR Green qPCR master mix (Bio-Rad). To normalize the data, *Puccinia sorghi* succinate dehydrogenase, cytochrome b subunit (KNZ46598.1) was chosen as an endogenous reference gene. In order to make the comparisons between samples, we made the assumption that this gene was expressed at similar levels across the two isolates, IN2 and IA16, in the susceptible maize line H95 plants. Three technical replicates of the RT-qPCR assay were used for each sample. Each biological replicate was sampled from three individual plants. The transcript expression level was calculated using the $2-\Delta\Delta CT$ method. The experiment was repeated two independent times. Similar results were obtained in the second repeat. Although they showed a similar pattern, one biological repeat was used for a graph due to a variation by uneven infection. The primers used for RT-qPCR are as follows (5′ to 3′): Pssdhc-1F_112 (CTTCCTTCGTACCTCGCTC); Pssdhc-1R_112 (GCTCGCAAACCTTTGT TGA); AvrRp1-D-qRT-F_162 (GACTGGCACAATGACATT); AvrRp1-D-qRT-R_162 (CT ATCGAGTTGAATTTTTGAGTG).

## Transient expression assay

Transient overexpression was assessed by the following procedure. *A. tumefaciens* carrying the various constructs were grown in LB medium overnight at 30°C with shaking, were collected by centrifugation (at $1,301 \times g$ in 16°C for 10 min) and were resuspended in induction buffer (10 mM $MgCl_2$, 10 mM MES [pH 5.6], and 200 mM acetosyringone). After incubation at 25°C for 2 h, *N. benthamiana* leaves were agroinfiltrated (OD 600 nm [$A_{600}$] = 0.3 or 0.6 for effectors and *Rp1* alleles including *Rp1-D*, respectively) and were harvested 2 to 4 dpi. Empty vector or Rp1-D21 was infiltrated at $A_{600}$ = 0.6 into one *N. benthamiana* leaf as a negative or positive control, respectively. All experiments were performed with three replicates.

## Transient expression of effectors with Rp1-D in *N. benthamiana*

251 effector candidates were amplified with primers that included 17 and 15-nucleotide tails: forward, 5'-CCAATCCCTCTACG-ATG-gene-specific sequence-3' and reverse, 5'-TATCC TCCTACG-stop codon-gene specific sequence-3' for use in the ligation-independent cloning (LIC) method [62]. To construct expressing cDNA clones, total RNA was extracted from the leaves of W22 infected with *P. sorghi* IN2 using TRIzol reagent. cDNA was synthesized from a total RNA template (3 μg) using Superscript III reverse transcriptase (Invitrogen). All amplified PCR products were cloned by the LIC method into a binary PVX-based pKW-LIC vector containing CaMV35S promoter and the NOS terminator cassette [63]. Each cloned vector was transformed into *A. tumefaciens* strain GV3101 for transient *in planta* expression assays. Both pKW-LIC and pKW-INF1 were used as negative and positive controls, respectively. Recombinant *Agrobacterium* carrying pKW-effectors were incubated at 28°C on an LB agar plate containing kanamycin and rifampicin for 1 day. Using toothpicks, the *Agrobacterium* was inoculated by piercing 5-week-old *N. benthamiana* leaves in which the *Rp1-D* gene was transiently expressed one day earlier. All experiments were performed with three replicates.

## Cloning of Rp1 alleles and AvrRp1-D deletion constructs

For cloning five *Rp1* alleles, total RNA was extracted from H95:Rp1-D using a TRIzol reagent by the manufacturer's protocol. cDNA was synthesized from a total RNA template (3 μg) using Superscript III reverse transcriptase. Five *Rp1* alleles and thirty effector candidates were cloned into the D-topo vector for gateway cloning. After sequencing, each clone was transferred into gateway vectors by LR reactions: pGWB2 (no tag), pGWB14 (with a 3xHA epitope in the C-terminus), pGWB617 (with a 4xcMYC epitope in the C-terminus) [64]. The ten *AvrRp1-D.1* deletion constructs were amplified using appropriate primer pairs (S4 Table). The resulting PCR products were cloned into D-topo and sequenced, then transferred into pGWB617 and pGWB641 (with an EYFP in the C-terminus) by gateway LR reaction.

## Maize protoplast cell-death assay

Ten seeds of H95 were germinated under light for 3 days (a 16h/8h photoperiod cycle, at day/night temperatures of 25/22 and relative humidity of 60% in growth chamber) and moved to a dark chamber for 7–8 days until the second leaf was about 8–15 cm. Etiolated leaves were cut to obtain the middle part of the second leaves using a fresh razor blade. 0.5 mm strips were cut and submerged under the enzyme solution (0.6 M mannitol, 10mM KCl, 10mM MES, 10mM $CaCl_2$, 1.5% cellulase, and 0.7% macerozyme). A vacuum was applied for 30 minutes for infiltration. Then, the mixture was incubated for another 2.5 hours with gentle shaking at 50 rpm in the dark. The protoplasts were finally released by shaking at 100 rpm for 30 minutes. The protoplasts were filtered with a 40 μm cell strainer. After filtering, the W5 buffer (154 mM

NaCl, 125 mM CaCl$_2$, 5 mM KCl, and 2 mM MES) was added to the tubes and re-suspended. They were centrifuged at 200 x g for 3 minutes, and the supernatant was removed. 3 mL of MMG buffer (0.4 M mannitol, 15 mM MgCl$_2$, and 4 mM MES) was added to the pellet and re-suspended by gentle shaking.

A hemocytometer was used to adjust the protoplast concentration to 1 x 10$^5$ /ml to transform the protoplast. Six µL of the LUC reporter construct, 14 µL of the Rp1-D construct, and 10 µL of the EV or AvrRp1-D construct (LUC/NLR/Avr ration = 3:7:5) were mixed for each transfection. A total of 30 µL of 10 µg plasmid mixture was added into a low protein binding tube and 300 µL of MMG buffer containing protoplasts. Immediately, 40% of PEG solution (40% PEG 4000, 0,4 mM mannitol, and 100 mM CaCl$_2$) was added to the tubes and mixed by 12 times horizontally inverting. They were incubated for one hour in the dark, and 2X 660 µl of W5 buffer was added to the tubes and mixed 8 times horizontally inverting. They were centrifuged at 200 x g for 3 minutes and removed supernatant. 960 µl of W5 buffer was added and incubated for 20–22 hours at 20/20˚C under the 16h/8h photoperiod. They were centrifuged for 3 minutes at 1000 x g to collect the protoplasts and removed the supernatant. For the luciferase assay, the lysis buffer (Promega, G7570) was added to the tubes containing the pellets, and the mixtures were incubated at room temperature for 10 minutes. 50 µl of each sample were aliquoted and transferred to 96 wells. 50 µl of luciferase substrate (Promega, G7570) was added into each well, and the signal was detected by Promega GloMax-96. All experiments were performed with five replicates.

## Protein blotting

For protein expression analysis, leaf tissue was collected 30–36 hpi. The samples were ground with pestles in liquid nitrogen, and the total protein was extracted in 200 µL Laemmle buffer (Bio-Rad). Samples were centrifuged at 14,000 g for 15 min at 25˚C, and 10 µL supernatants were loaded for SDS-PAGE. Proteins were transferred to nitrocellulose membrane (GM) and analyzed by western blot. HA detection was performed using a 1:500 dilution of anti-HA-HRP (horseradish peroxidase) (Cat# 12013819001, Roche). The cMYC detection was performed using a 1:5000 dilution of anti-cMYC-HRP (Cat# R95125, Invitrogen). The HRP signal was detected by the ECL substrate kit (Supersignal west femto chemiluminescent substrate, Thermo Scientific).

For yeast protein expression analysis, yeast cells were grown from a colony in 50 ml of DDO (SD-L-W) at 30˚C for 2 days. After harvesting the cells by centrifugation, they were resuspended in 200 µl of 1X urea Laemmli buffer (50 mM Tris-HCl pH 6.8, 8 M urea, 1% β-mercaptoethanol, 2 mM EDTA, 5% glycerol, 0.004% bromophenol blue, and 1X Protease Inhibitor Cocktail (Roche)). The cells were lysed by repetitive freezing and thawing five times, each for two minutes. The yeast suspension was transferred into Protein LoBind tubes with glass beads and homogenized 10 times at 30 Hz for 30 seconds, with a 15-second break between each cycle, using a Tissuelyser II (Qiagen). 2% SDS was added to the crude protein extract, which was then denatured by boiling for 15 minutes. The proteins were separated on an 8% SDS-PAGE gel and transferred onto a PVDF membrane for immunoblotting (Bio-Rad). AD- or BD-binding proteins were detected using anti-GAL4 (Abcam, Cat No. ab135398, 1:2000 dilution) and anti-GAL4 (DBD) (Santa Cruz Biotechnology, Cat No. sc-510, 1:1000 dilution), respectively.

## Yeast two-hybrid assay

To generate a yeast expression vector with *Rp1* alleles, their deletion constructs, and effector, *Rp1* alleles and their deletion constructs were cloned into a yeast expression vector containing

GAL4 activation domain (addgene #20161), and the effector gene was cloned the vector containing GAL4 binding domain (addgene #20162). Detailed information was listed in S5 and S6 Tables. All constructs were generated by gateway compatible cloning method. Protein interaction in the yeast was confirmed by following the manufacturer protocol (Takara, Matchmaker yeast-two hybrid system). Briefly, after mating haploid yeast, each diploid yeast was grown in the synthetic defined dropout media without leucine and tryptophan for 2 days. After growing the selected yeast in yeast, peptone dextrose adenine (YPDA) liquid media, an optical density of each diploid yeast was adjusted as 0.5 and 10 μl of yeast suspension was dropped on the synthetic dropout media without leucine, tryptophan and histidine until yeast growth was observed. Experiments were repeated three times.

## Chitin-mediated ROS burst assay

Reactive oxygen species were detected using a luminol-based chemiluminescence assay as previously described with slight modifications [65, 66]. *A. tumefaciens* GV3101 carrying either AvrRp1-D:4XcMYC or free sYFP2 [67] were infiltrated ($OD_{600}$ = 0.5) into 3-to-5-week-old *N. benthamiana* leaves. Forty-eight hours following agroinfiltration, leaf discs (5 mm diameter) were harvested using a cork borer, washed in deionized water, and incubated overnight in sterile water in a 96-well OptiPlate microplate (Perkin Elmer). The following day, the sterile water was removed and replaced with a chitin elicitation solution (luminol (30 μg/mL), horseradish peroxidase (20 μg/mL), chitin [hexamer] (5 μg/mL), and sterile water). ROS production and accumulation were monitored by luminescence for 40 minutes using a microplate reader (Tecan Infinite M200 Pro plate reader).

## Confocal microscopy

The abaxial sides of *N. benthamiana* leaves infiltrated by agrobacteria were cut for observation at 30 hpi for constructs inducing HR by a Zeiss LSM 880 Axio Examiner confocal microscope with a Plan Apochromat 20x/0.8 objective. YFP fluorescence was excited at 488 nm and was observed between 495 and 550 nm. TaqRFP was excited at 561 nm and was observed between 580 and 675 nm.

## Supporting information

**S1 Fig.** PCA plot of 72,538 *P. sorghi* transcripts expressed in H95, H95:Rp1-D, and in haustoria at different time points.
(TIF)

**S2 Fig. Toothpick assay in *N. benthamiana*.** (A) and (B): Rp1-D was transiently expressed in the entire *N. benthamiana* leaf by agro-infiltration using a needle-less 1ml syringe and ten effector candidates expressed in a Potato Virus X-based pKW-LIC vector were then toothpicked into the leaf in a row of four 1 day afterward. pKW-INF1 was used as a positive control to induce cell death. pKW-ΔGFP was used as a negative control. A representative photo was taken at 5 dpi. The experiments were repeated three times with the same results. Three leaves for biological repeat were tested in each experiment.
(TIF)

**S3 Fig. Transient expression of 4A12 and 597 in *N. benthamiana*.** Rp1-D21:3xHA was used as a positive control. A representative photo was taken at 4 dpi. The dashed circles indicate areas of infiltration. 9 individual plants were infiltrated and showed similar results.
(TIF)

**S4 Fig. Structure of AvrRp1-D.1 and AvrRp1-D.2 and functional study of AvrRp1-D variants with Rp1-D *in N. benthamiana*.** (A) An alignment of AvrRp1-D.1 and AvrRp1-D.2 amino acids. AvrRp1-D.2 has a 21 amino acid deletion in the ORF. (B) Co-expression of Rp1-D (fused with a C-terminal tag 3xHA) with AvrRp1-D.1 or AvrRp1-D.2 (fused with a C-terminal 4xcMYC). Representative leaf was photographed at 5 dpi. 9 individual plants were infiltrated and showed similar results. (C) Protein expression of Rp1-D and AvrRp1-D.1 variants transiently expressed in *N. benthamiana*. Total protein was extracted from agro-infiltrated leaves at 36 hpi, and anti-HA or anti-cMYC antibody was used to detect the expression of the fused proteins. The sizes of the proteins are indicated on the right. Ponceau S staining of the Rubisco subunit showed equal loading of protein samples. Three independent biological replicates were tested, and they showed similar results.
(TIF)

**S5 Fig. Co-expression of Rp1-D and AvrRp1-D.1 with signal peptide.** Co-expression of Rp1-D with AvrRp1-D.1 with signal peptide does not induce observable cell death in *N. benthamiana*. 9 individual plants were infiltrated and showed similar results. Empty vector (EV) was used as a negative control. Rp1-D21 expression was used as the positive control.
(TIF)

**S6 Fig. AvrRp1-D.2 effector induces weaker cell death with Rp1-dp2.** (A) Rp1-dp1, -dp2, -dp5, -dp6, -dp7, and -dp8 fused with a C-terminal tag 3xHA were co-infiltrated with AvrRp1-D.2 fused with a C-terminal 4xcMYC tag or EV in *N. benthamiana*. Representative photos were taken at 4 dpi. 9 individual plants were infiltrated and showed similar results. (B) Protein expression of Rp1 alleles and AvrRp1-D.2. Total protein was extracted from agro-infiltrated leaves at 36 hpi, and anti-HA or anti-cMYC antibody was used to detect the expression of the fused proteins. The sizes of the proteins were labeled on the right. Ponceau S staining of the Rubisco subunit showed comparative levels of protein samples in each lane.
(TIF)

**S7 Fig. Alignment of Rp1-D, Rp1-dp2, and Rp1-dp7 amino acid sequences.** The CC, NB-ARC and LRR domains are indicated with purple, red and blue letters respectively. Black, green, and red lines indicate the LRR1, LRR2 and LRR3 regions shown. A green box indicates MHD and LHD motifs. All shades in different color indicate the difference among three alleles; Magenta: Rp1-D and -dp2 vs. -dp7, Green: Rp1-D and -dp7 vs. -dp2, Cyan: Rp1-D vs. -dp2 and -dp7, Purple: all three.
(TIF)

**S8 Fig. Co-expression of Rp1 mutants with AvrRp1-D.1 or AvrRp1-D.2 in *N. benthamiana*.** (A) All constructs tested were described previously (5). Rp1-D21 was used as a positive control. 9 individual plants were infiltrated and showed similar results. (B) The Rp1 allele mutants were fused with the C-terminal 3xHA and AvrRp1-D.1 and AvrRp1-D.2 were fused with the C-terminal 4xcMYC. Total protein was extracted from agro-infiltrated leaves at 36 hpi, and anti-HA or anti-cMYC antibody was used to detect the expression of the fused proteins. The sizes of the proteins were labeled on the right. Ponceau S staining of the Rubisco subunit showed equal loading of protein samples.
(TIF)

**S9 Fig. Alignment of AvrRp1-D.1 identified from the avirulent isolate *P. sorghi* IN2 and AvrRp1-D[IA16] identified from the virulent isolate *P. sorghi* IA16.** Only one amino acid is different in the alignment. Amino acid 51 is Arginine in IN2 and lysine in IA16.
(TIF)

**S10 Fig. The expression level of *AvrRp1-D.1* and *AvrRp1-D.2* in the susceptible maize line H95 plants.** Expression level of *AvrRp1-D.1* and *AvrRp1-D.2* from the avirulent *P. sorghi* IN2 and the virulent *P. sorghi* IA16 (paired *t*-test, *p<0.05, **p<0.01) in the second repeat. One biological repeat was used for the graph.
(TIF)

**S1 Table. Predicted secreted proteins from the *P. sorghi* IN2 haustorial transcriptome.**
(XLSX)

**S2 Table. The list of screened 251 effector candidates sequenced by whole plasmid sequencing.**
(XLSX)

**S3 Table. The nine Rp1-D paralogues at the *Rp1-D* locus in maize arranged according to their similarity to Rp1-D. Adapted from table 1 in Sun et al, 2001. (27).**
(XLSX)

**S4 Table. Primers used in this study to amplify *Rp1-D* paralogues and *AvrRp1-D* deletion constructs.**
(XLSX)

**S5 Table. Constructs used for yeast-two hybrid.**
(XLSX)

**S6 Table. Primers for yeast two hybrid.**
(XLSX)

**S1 Data. 72,538 *de novo* assembled sequences.**
(FASTA)

**S2 Data. Non-cropped western blot membranes.**
(PDF)

# Acknowledgments

We thank Dr. Mikyung Sung, Dr. Thi Kim Ha Nguyen, and Sangsik Yun for cloning genes and RT-qPCR, Dr. Christian Elowsky for confocal microscopy and Dr. Carole Saravitz, Dr. Joe Chiera, and the staff of the NCSU Phytotron for providing plant growth facilities and care for the plants. Dr. Shannon Sermons, Greg Marshall, and Reese G. Milburn provided valuable technical assistance.

All opinions expressed in this paper are the author's and do not necessarily reflect the policies and views of USDA. USDA is an equal opportunity provider and employer.

# Author Contributions

**Conceptualization:** Saet-Byul Kim, Peter Balint-Kurti.

**Data curation:** Saet-Byul Kim, Ki-Tae Kim, Gir-Won Lee.

**Formal analysis:** Saet-Byul Kim, Ki-Tae Kim, Namrata Jaiswal, Gir-Won Lee, Quentin D. Read.

**Funding acquisition:** Saet-Byul Kim, Matthew Helm, Ralph Dean, Eunsook Park, Peter Balint-Kurti.

**Investigation:** Saet-Byul Kim, Solhee In, Namrata Jaiswal, Seungmee Jung, Abigail Rogers, Libia F. Gómez-Trejo, Sujan Gautam, Hee-Kyung Ahn.

**Methodology:** Saet-Byul Kim, Ki-Tae Kim, Gir-Won Lee.

**Project administration:** Saet-Byul Kim.

**Resources:** Saet-Byul Kim, Katerina L. Holan, Steven A. Whitham, Eunsook Park, Peter Balint-Kurti.

**Software:** Ki-Tae Kim.

**Supervision:** Saet-Byul Kim, Ralph Dean, Eunsook Park.

**Validation:** Saet-Byul Kim.

**Visualization:** Saet-Byul Kim, Ki-Tae Kim, Solhee In, Namrata Jaiswal, Gir-Won Lee, Seungmee Jung, Abigail Rogers, Matthew Helm.

**Writing – original draft:** Saet-Byul Kim, Ki-Tae Kim, Solhee In, Seungmee Jung, Matthew Helm, Peter Balint-Kurti.

**Writing – review & editing:** Saet-Byul Kim, Libia F. Gómez-Trejo, Matthew Helm, Hee-Kyung Ahn, Hye-Young Lee, Jongchan Woo, Katerina L. Holan, Steven A. Whitham, Jonathan D. G. Jones, Doil Choi, Ralph Dean, Eunsook Park, Peter Balint-Kurti.

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
