## [Decision Letter · Decision Letter 0]

8 Apr 2024

Dear Dr. Kim,

Thank you very much for submitting your manuscript "Use of the Puccinia sorghi haustorial transcriptome to identify and characterize AvrRp1-D recognized by the maize Rp1-D resistance protein" for consideration at PLOS Pathogens. As with all papers reviewed by the journal, your manuscript was reviewed by members of the editorial board and by several independent reviewers. In light of the reviews (below this email), we would like to invite the resubmission of a significantly-revised version that takes into account the reviewers' comments.

Both reviewers suggest additional experiments to support the conclusions in this manuscript. It will be essential to address these concerns in a significantly revised version. In particular, please provide immunoblots for the Y2H experiments and clarify the role of the Cterminal LRR repeats in a set of swap experiments that complement the existing dataset (reviewer 1), provide expression data of the two Avr alleles in infected tissues by qRT-PCR or RNA-seq and compare the HR by Rp1-D/AvrRp1-D and Rp1-D/ AvrRp1-DIA16 in maize protoplasts (reviewer 2).

We cannot make any decision about publication until we have seen the revised manuscript and your response to the reviewers' comments. Your revised manuscript is also likely to be sent to reviewers for further evaluation.

Sincerely,

Sebastian Schornack, Ph.D.

Academic Editor

PLOS Pathogens

Bart Thomma

Section Editor

PLOS Pathogens

Michael Malim

Editor-in-Chief

PLOS Pathogens

orcid.org/0000-0002-7699-2064

Both reviewers suggest additional experiments to support the conclusions in this manuscript. It will be essential to address these concerns in a significantly revised version. In particular, please provide immunoblots for the Y2H experiments and clarify the role of the Cterminal LRR repeats in a set of swap experiments that complement the existing dataset (reviewer 1), provide expression data of the two Avr alleles in infected tissues by qRT-PCR or RNA-seq and compare the HR by Rp1-D/AvrRp1-D and Rp1-D/ AvrRp1-DIA16 in maize protoplasts (reviewer 2).

Reviewer's Responses to Questions

**Part I - Summary**

Reviewer #1: In their manuscript, Kim et al. identified and cloned AvrRp1-D, the avirulence gene from the biotrophic maize rust pathogen Puccinia sorghi. The authors used in silico prediction of effector genes based on haustorial and time-course transcriptomic resources obtained from infected maize plants. Through a transcriptome assembly derived from these resources, coupled with signal peptide predictions, expression analysis, and transmembrane domain prediction, they identified a set of highly expressed candidate effectors at early infection time points. Subsequently, they generated a screenable effector library based on viral vectors and identified two effectors that induced Rp1-D-dependent cell death in both N. benthamiana and maize protoplasts. The identified AvrRp1-D stands out as an unusual effector due to its size, which exceeds the typical range observed for Avrs from other rusts and plant pathogens. Through a series of truncation experiments, the authors identify the N-terminal part of the Avr protein as responsible for recognition by the Rp1-D protein. Furthermore, the authors presented evidence suggesting that AvrRp1-D might play a role in suppressing chitin-mediated reactive oxygen species (ROS) accumulation in the heterologous system of N. benthamiana. Lastly, through Y2H and domain swap experiments between Rp1-D and a weaker allele Rp1 allele, the authors provide evidence for the C-terminal part of the LRR domain being involved in the activation of Rp1-D by AvrRp1-D.

The transcriptomic-based approach used by the authors to identify potential candidate effectors seems a promising approach for other pathosystems for which large-scale genomic resources are still missing. In addition, the finding that commonly used criteria for effector definition in fungi (especially in regards to size) might be too stringent also represents a highly relevant finding. Therefore, this manuscript is of broad interest to the field.

While throughout the manuscript experiments are well-designed and contain appropriate controls, I have the following reservations concerning the authors' conclusion that the C-terminal part LRR domain is the point of interaction with the AVR. Prior to publication, the point below should be addressed:

Reviewer #2: The authors identified the putative cognate effector recognized by Rp1-D by transiently co-express Rp1-D with a list of predicted P. sorghi effectors in N. benthaminan. The AvrRp1-D is an unusually large avirulence gene and can cause cell death when co-expressed with Rp1-D in both N. benthaminan and maize protoplasts. Rp1-D and Avr-Rp1D physically interact with each other. The authors also characterized regions in Rp1-D and AvrRp1-D important for the recognition/activation. In summary, the authors have provided strong evidence for the identification of the first R-Avr pair for common rust in maize. However, the authors also describe observations which aren’t consistent with the conclusion.

**Part II – Major Issues: Key Experiments Required for Acceptance**

Reviewer #1: Yeast-to-hybrid assay: The authors should provide western blots of the fusion proteins used in the Y2H assay to show that they are indeed produced in yeast. Otherwise, absence of interaction cannot be concluded from the data provided in the manuscript.

Lines 201-206: I have some reservations regarding the interpretation of the results in Figure 4(A). While the V17 construct, containing only the very C-terminal LRR repeats of Rp1-dp2, loses the ability to induce a hypersensitive response (HR), it's noteworthy that when a larger portion of the LRR repeat from Rp1-dp2 (V16) is introduced into Rp1-D, this chimeric construct displays a stronger HR reaction compared to the original Rp1-dp2. This observation suggests that not only the C-terminal part but also the additional portions of the LRR repeat are vital for the HR response. It implies that the presence of the C-terminal LRR in Rp1-D may not be strictly necessary for HR. Therefore, I suggest clarifying the role of the Cterminal LRR repeats in a set of swap experiments that complement the existing dataset with a swapping partner that, in contrast to Rp1-dp2, does not induce any HR response upon AvrRp1-D recognition (such as the Rp1-d7 allele also used in the Y2H assays).

Reviewer #2: I have two suggestions for the authors to consider:

1. The authors described a modified protocol on haustoria isolation, which enables them to generate a haustorial transcriptome assembly and a list of predicted P. sorghi effectors. Presumably, this is a significant technical improvement for effector discovery. Since the authors have also generated RNA-seq data from infected tissues at different time points, it’d be very beneficial to run a parallel effector prediction with the RNA-seq data sets and compare the predicted effector lists from haustoria and infected tissues. It’d be interesting to understand the extent of improvement (e.g., how many additional effectors were predicted from the haustorial transcriptome assembly?).

2. The authors identified an AvrRp1-D allele from a virulent isolate IA16, and the AvrRp1-DIA16 unexpectedly induced Rp1-D-dependent HR in N. benthamiana. The observation seems in contrary to the conclusion that this is the authentic Avr for Rp1-D. The authors speculated several hypotheses to explain the conflicts, but the most plausible explanation is expression variations between the two Avr alleles in infected tissues, which should be easy to compare by qRT-PCR or RNA-seq. I hope the authors can generate such expression data. Also, while Rp1-D + AvrRp1-D induces strong HR in N. benthamiana (comparable to Rp1-D21), Rp1-D + AvrRp1-D only gives a weak HR in maize protoplasts (much weaker than Rp1-D21, Fig. 2D). This raises the possibility that HR by Rp1-D/AvrRp1-D may not be strong enough to fully explain the resistance from Rp21 in maize (HR is “over-estimated” in N. benthamiana). The author may compare HR by Rp1-D/AvrRp1-D and Rp1-D/ AvrRp1-DIA16 in maize protoplasts. If Rp1-D/ AvrRp1-DIA16 gives a much weaker HR than Rp1-D/AvrRp1-D in maize, it’d support the authors’ main conclusion (AvrRp1-D is the Avr for Rp1-D). Otherwise, the authors should consider not to make such a strong conclusion (e.g., may call it a putative AvrRp1-D).

**Part III – Minor Issues: Editorial and Data Presentation Modifications**

Reviewer #1: Figure 1 panel D, what does relative expression level mean? To what are the read counts normalize? Please specifiy.

Figure 4, panel A: AvrRp1-D with construct Rp1-D21 is indicated as +++ but there is no HR visible on the picture

Figure 1, panel D, was the empty vector only transfected once? Because the individual datapoints indicated values for biological replicates are not shown.

Methods Line 386-398, Please provide more detail in the description of the in-silico predictions utilized in this study. Specify the thresholds applied in the different prediction tools such as SignalP and EffectorP etc. . Additionally, provide details how this information was utilized to predict and prioritize the candidate effector list for experimentation. Furthermore, provide details regarding the threshold employed to identify homologs against maize and specify the source of the reference maize genome utilized

Methods Line 409: Please specify the primer sequences that you used to create the effector library using LIC cloning.

Method Line: 443, where the plasmids containing R and AVR or EV plasmides mixed in equimolar ratio? Please specify.

2) Transcirptome assembly: the transcriptome assembly should be provided in a readily accessible format, such as a multifasta file (machine readable format) on a repository (Zenodo for instance) instead of providing it as part of an excel file.

Discussion:

Line 283-285, did you provide this information somewhere in the results and method section? Please specify what you mean by 94%? Where 94% of transcripts identical between the two methods?

Line 301-304, here it would be interesting if you discuss your own findings a bit more carefully. Your finding that the NLS signal is not required for Rp1-D activation supports a cytoplasmic NLR activation, or at least it does not contradict it.

Line 325: “data not shown”, is not accepted anymore. If you have, the data why not show it for instance in a supplementary figure.

1) Throughout the manuscript: uncropped western blots should be provided as supplementary material.

Supporting Figure 4, panel C, loading control for one of the western blots is missing.

Line 119, for a better understanding of the experimental workflow; please specify how the information of the Signalp4.1 and TmHmm prediction were used to filter the transcripts.

Reviewer #2: (No Response)

PLOS authors have the option to publish the peer review history of their article (what does this mean?). If published, this will include your full peer review and any attached files.

Reviewer #1: No

Reviewer #2: No
---

## [Decision Letter · Decision Letter 1]

11 Oct 2024

Dear Dr. Kim,

We are pleased to inform you that your manuscript 'Use of the Puccinia sorghi haustorial transcriptome to identify and characterize AvrRp1-D recognized by the maize Rp1-D resistance protein' has been provisionally accepted for publication in PLOS Pathogens.

Before your manuscript can be formally accepted you will need to complete some formatting changes, which you will receive in a follow up email. A member of our team will be in touch with a set of requests. Please also use this opportunity to correct the number mistake raised by one reviewer.

Best regards,

Sebastian Schornack, Ph.D.

Academic Editor

PLOS Pathogens

Bart Thomma

Section Editor

PLOS Pathogens

Michael Malim

Editor-in-Chief

PLOS Pathogens

orcid.org/0000-0002-7699-2064

Reviewer Comments (if any, and for reference):

Reviewer's Responses to Questions

**Part I - Summary**

Reviewer #1: The authors have made significant improvements to the revised manuscript, and my previous concerns have been addressed. Specifically:

i) They have provided Western blot analyses demonstrating that the proteins used in the Y2H assay were effectively produced, which substantially strengthens their conclusion that certain proteins do not interact with the identified AVR.

ii) They have included additional NLR-chimera experiments and appropriately adjusted their claims regarding the role of the last part of the LRR domain in AVR recognition.

iii) They have expanded the Materials and Methods section with important details on the in silico analyses, enhancing the reproducibility of their approach.

I therefore recommend accepting the manuscript in its current form.

Reviewer #2: The authors have satisfactorily addressed the recommendations. They demonstrated a clear enrichment of fungal transcripts in haustoria compared to infected tissues. More importantly, they revealed significant expression variation between the virulent and avirulent AvrRp1-D21 alleles, providing a plausible explanation for the unexpected induction of Rp1-D-dependent HR by AvrRp1-DIA16.

**Part II – Major Issues: Key Experiments Required for Acceptance**

Reviewer #1: (No Response)

Reviewer #2: (No Response)

**Part III – Minor Issues: Editorial and Data Presentation Modifications**

Reviewer #1: (No Response)

Reviewer #2: On Line 139, it should be ‘72,538 fungal genes”, not “72,538 effector genes”.

PLOS authors have the option to publish the peer review history of their article (what does this mean?). If published, this will include your full peer review and any attached files.

Reviewer #1: No

Reviewer #2: No

---

## [Editor Report · Acceptance letter]

4 Nov 2024

Dear Dr. Kim,

We are delighted to inform you that your manuscript, "Use of the Puccinia sorghi haustorial transcriptome to identify and characterize AvrRp1-D recognized by the maize Rp1-D resistance protein," has been formally accepted for publication in PLOS Pathogens.

Best regards,

Michael Malim

Editor-in-Chief

PLOS Pathogens

orcid.org/0000-0002-7699-2064